# A Fully Integrated AC-DC Converter in 1 V CMOS for Electrostatic Vibration Energy Transducer with an Open Circuit Voltage of 10 V

**Yosuke Ishida** † **and Toru Tanzawa** *

Graduate School of Integrated Science and Technology, Shizuoka University, Hamamatsu 432-8561, Japan;
ishida.yohsuke.15@shizuoka.ac.jp
* Correspondence: toru.tanzawa@shizuoka.ac.jp
† Current address: Anritsu Corp., Atsugi 243-8555, Japan.

**Abstract:** This paper proposes an AC-DC converter for electrostatic vibration energy harvesting. The converter is composed of a CMOS full bridge rectifier and a CMOS shunt regulator. Even with 1 V CMOS, the open circuit voltage of the energy transducer can be as high as 10 V and beyond. Bandgap reference (BGR) inputs a regulated voltage, which is controlled by the output voltage of the BGR. Built-in power-on reset is introduced, which can minimize the silicon area and power to function normally found upon start-up. The AC-DC converter was fabricated with a 65 nm low-Vt 1 V CMOS with 0.081 mm². 1 V regulation was measured successfully at 20–70 °C with a power conversion efficiency of 43%.

**Keywords:** AC-DC converter; shunt regulator; full bridge rectifier; electrostatic vibration energy harvesting; fully integrated; IoT

## 1. Introduction

Energy harvesting (EH) is technology for harvesting power for IoT edge devices from environmental energy using energy transducers [1]. Eliminating the replacement of batteries based on EH can reduce the total cost of IoT devices. Electrostatic energy transducers (ES-ETs) can convert vibration energy into electronic power [2,3]. Due to the high output impedance of 1 MΩ or larger, open circuit voltages have to go beyond 10 V to generate power of 10 μW or larger. In [4], a battery charger is proposed using two variable capacitors based on ES-ETs. Capacitance varies with vibration, resulting in variable voltage at the capacitor node. Two capacitors vary out of phase. Thus, the diode connected with those two capacitors flows current from one to the other. The latter capacitor is connected to the battery via another diode. As a result, with sufficient amplitude in the voltage at the capacitor node, the battery can be charged with vibration energy. Another power converter is proposed in [5,6] based on a full bridge rectifier (FBR) followed by a DC-DC buck converter, as shown in Figure 1a. An HV rectifier is composed of four diodes for converting the AC power of ES-ETs into DC power to the converter. As the DC voltage is much higher than the maximum voltage acceptable to sensor CMOS ICs, power management circuits (PMC) in DC/DC converters need to be fabricated using a BCD process, which provides an HV CMOS operating even at high voltages of 10 V or higher. Buck converters require external components such as inductors, capacitors, and resistor (LCRs) to convert the DC input voltage of an order of 10 V into an output voltage of an order of 1 V. The priority in design was power conversion efficiency at a power of 1 mW rather than the cost and form factor. What if the priority should be the cost and form factor? In this work, we focus on the full integration of a converter into the same chip for a sensor/RF, as shown in Figure 1b. Section 2 discusses key design features. Power-on-reset (POR) is a critical block to starting up the operation. A built-in POR with no additional power is proposed. The circuit was

fabricated in 65 nm low-Vt CMOS. Experimental results are shown in Section 3. Section 4 compares the circuit features of the proposed circuit with previously reported converters.

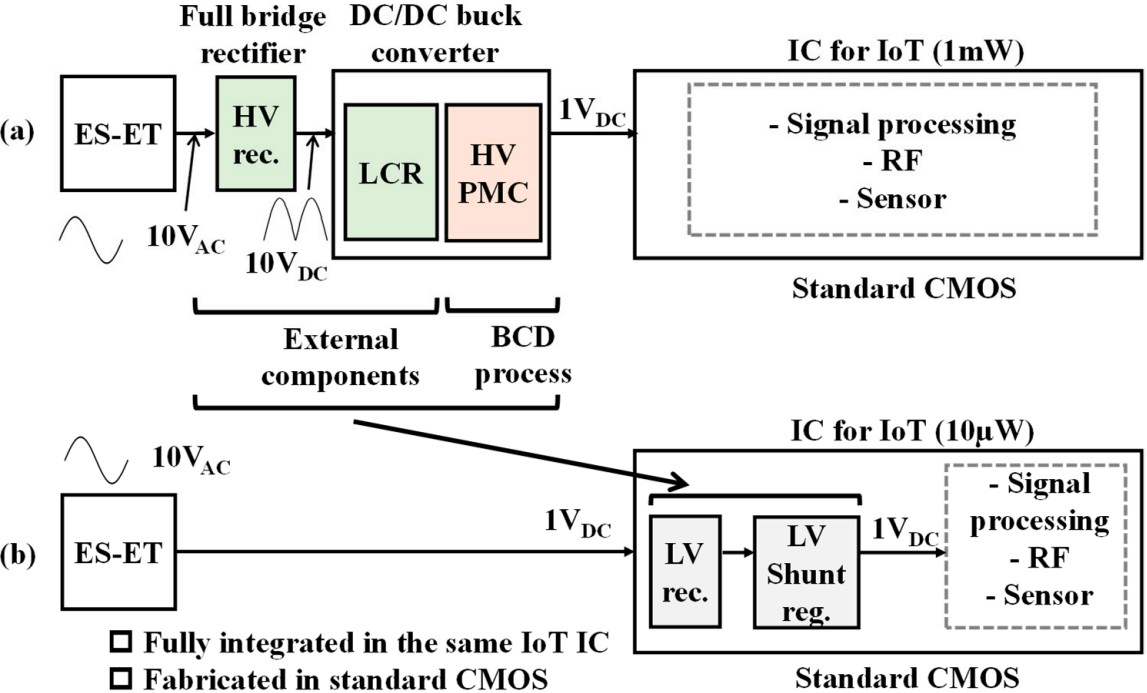

**Figure 1.** IoT edge device with an ES-ET and (**a**) a buck converter with FBR and PMC in a BCD process [6] or (**b**) a proposed shunt regulator integrated into an IoT chip.

## 2. Circuit Design

### 2.1. System Design

Figure 2 illustrates a proposed fully integrated AC-DC converter. A cross-coupled CMOS bridge circuit [7] is used in a full bridge rectifier (FBR). An additional diode-connected NMOS ($N_D$) is needed in order to not flow reverse current when the voltages at IN1 and IN2 become lower than the regulated output voltage $V_{DD}$. An active diode [8] can be placed in parallel with $N_D$ to reduce the voltage drop. In this design, low power is prioritized. An active diode requires an opamp, which consumes power. Bandgap reference (BGR) operates with $V_{DD}$, which is controlled by the output voltage $V_{REF}$.

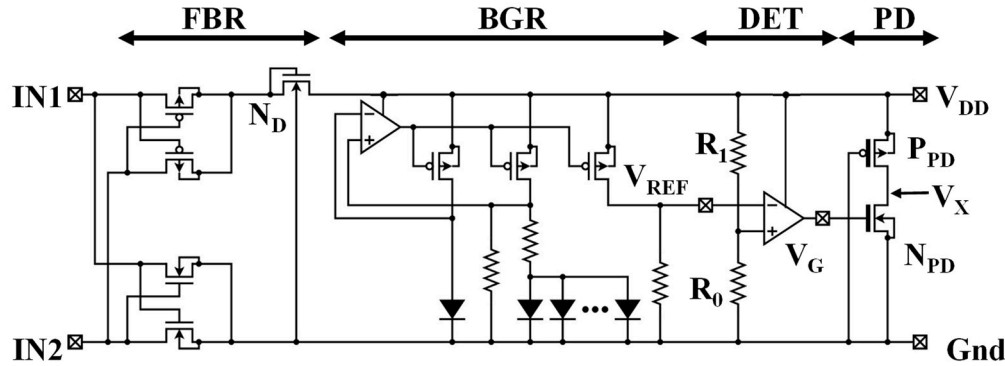

**Figure 2.** AC-DC converter for an ES-EH proposed in this work.

### 2.2. Power Conversion Efficiency

Figure 3 is used to estimate the power conversion efficiency (PCE). When the voltage drop at a rectifier is sufficiently low in comparison with the amplitude of the voltage source ($V_A$) and the DC output ($V_{DD}$), an average output power can be estimated by (1) in a steady state. The maximum available power $P_{AV}$, which is defined by the output power when the load resistance is as large as the output impedance of the transducer $|Z_S|$, is given by (2). Then, PCE is calculated by (3). Figure 4 shows η vs. $V_{DD}$ at $V_A$ = 10 V, 30 V, 60 V, and 100 V. When $V_{DD}$ is controlled to be 1 V, η decreases as $V_A$ (>10 V) increases, which is the weakest point for the proposed circuit. Lower $V_A$, or, in other words, lower output impedance, is preferred for an electrostatic energy transducer.

$$\overline{P_{OUT}} = \frac{2V_{OUT}^2}{\pi|Z_S|}\left\{ \sqrt{\left(\frac{V_m}{V_{OUT}}\right)^2 - 1} - \cos^{-1}\left(\frac{V_{OUT}}{V_m}\right)\right\} \tag{1}$$

$$\overline{P_{AV}} = \frac{V_m^2}{8|Z_S|} \tag{2}$$

$$\eta = \frac{\overline{P_{OUT}}}{\overline{P_{AV}}} \tag{3}$$

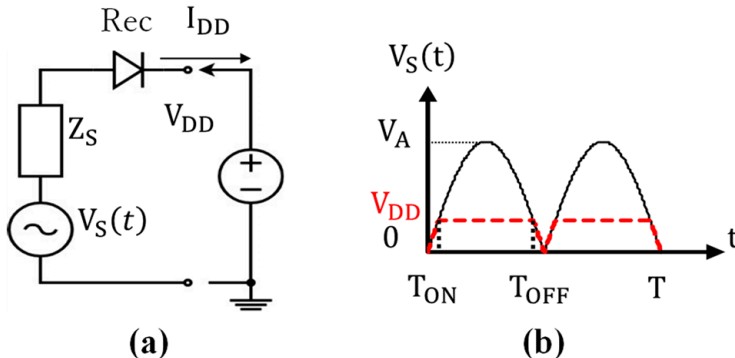

**Figure 3.** Circuit model for estimating the power conversion efficiency. (**a**) Circuit model; (**b**) voltage waveform.

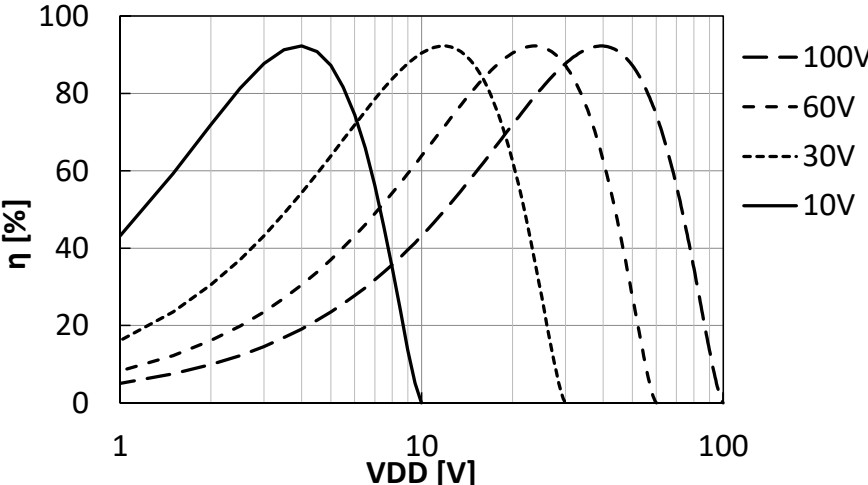

**Figure 4.** PCE η vs. $V_{DD}$ at $V_A$ = 10 V, 30 V, 60 V, and 100 V.

### 2.3. Bandgap Reference (BGR)

As the target $V_{DD}$ is 1 V, a current-mode bandgap reference (BGR) [9] was selected in this work. Using low-Vt CMOS, the reference voltage $V_{REF}$ is stable at 0.8 V and higher, as shown in Figure 5a. Operation current $I_{DD}$, including a current generator, is about 200 nA at $V_{DD}$ = 1 V, as shown in Figure 5b. The detector (DET) controls $V_{DD}$ to be 2X $V_{REF}$ with $R_0 = R_1$ (see Figure 2). Table 1 summarizes the simulated results of the AC-DC converter at corner conditions.

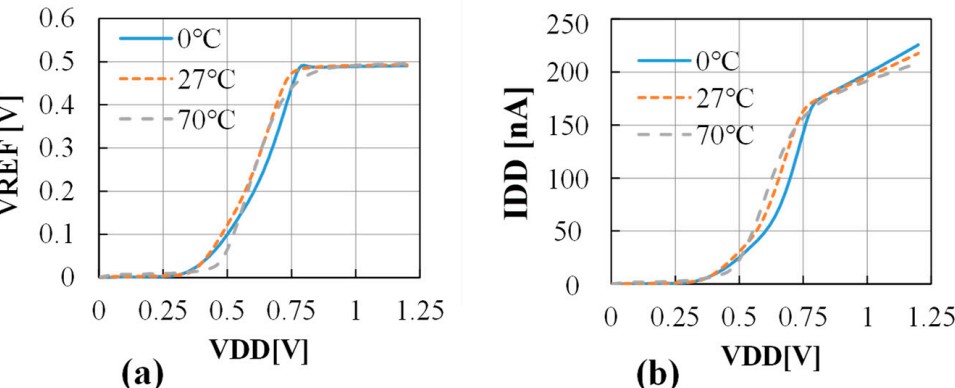

**Figure 5.** (**a**) Simulated $V_{REF}$ vs. $V_{DD}$ and (**b**) $I_{DD}$ vs. $V_{DD}$.

**Table 1.** Simulation result of $V_{DD}$ at corner conditions.

| VDD[V] | FF | TT | SS |
|---|---|---|---|
| 0 °C | 1.0 | 1.0 | 1.0 |
| 27 °C | 1.0 | 1.0 | 1.0 |
| 70 °C | 0.90 | 1.0 | 1.0 |

### 2.4. Built-In Power-On Reset (POR)

After the transducer starts generating power, $V_{DD}$ is gradually increased from 0 V. Every building block has its own minimum operating $V_{DD}$. If pull-down (PD) is enabled below the minimum $V_{DD}$, the system is latched in that state and therefore $V_{DD}$ should no longer be increased. Power-on reset (POR) aims to remove such misbehavior. Additional low-power POR requires more silicon area and more power. As a result, we simply added a blocking PMOS ($P_{PD}$), which has a standard threshold voltage in the pull-down path, as shown in Figure 2. As shown in Figure 6a, while $V_{DD}$ is low, $V_{REF}$ can be also lower than $V_{DD}/2$ because of the misbehavior of the OPAMPs. In that case, the gate voltage $V_G$ of the pull-down NMOS ($N_{PD}$) stays high. Even in such a case, $P_{PD}$ disconnects the path from $V_{DD}$ to ground. The necessary condition for normal operation is that $P_{PD}$ starts conducting after $V_{REF} > V_{DD}/2$. Figure 6b shows the simulated waveform of the entire system to verify the normal operation during power-up.

### 2.5. Decoupling Capacitor

To stabilize $V_{DD}$ even with AC input, a decoupling capacitor $C_{VDD}$ needs to be placed. When a maximum output current of 10 μA at AC power frequency of 100 Hz is needed, the capacitance of $C_{VDD}$ is required to be 10 μA X 5 ms/10 mV~5 μF for a ripple in $V_{DD}$ of 10 mV. The entire AC-DC converter is simulated in AC mode, as well as with different capacitance values of $C_{VDD}$, as shown in Figure 7. The system can be stable with $C_{VDD}$ of 20 nF or larger.

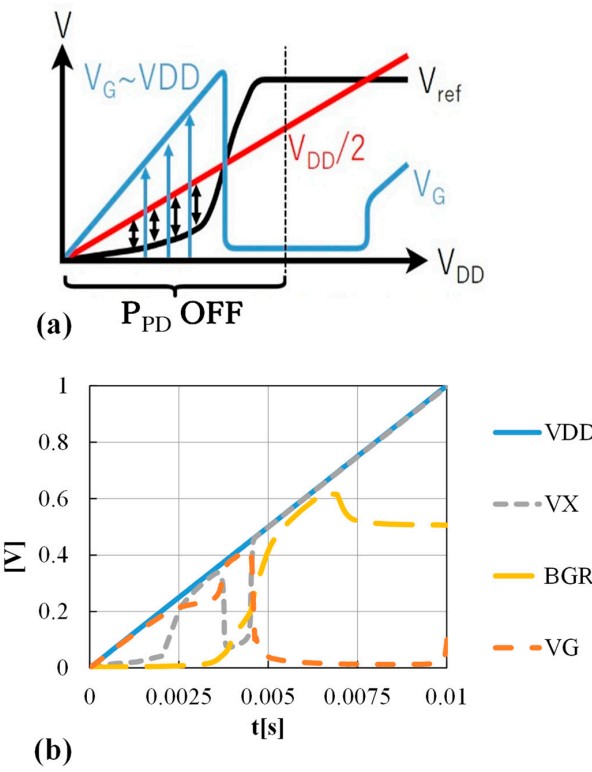

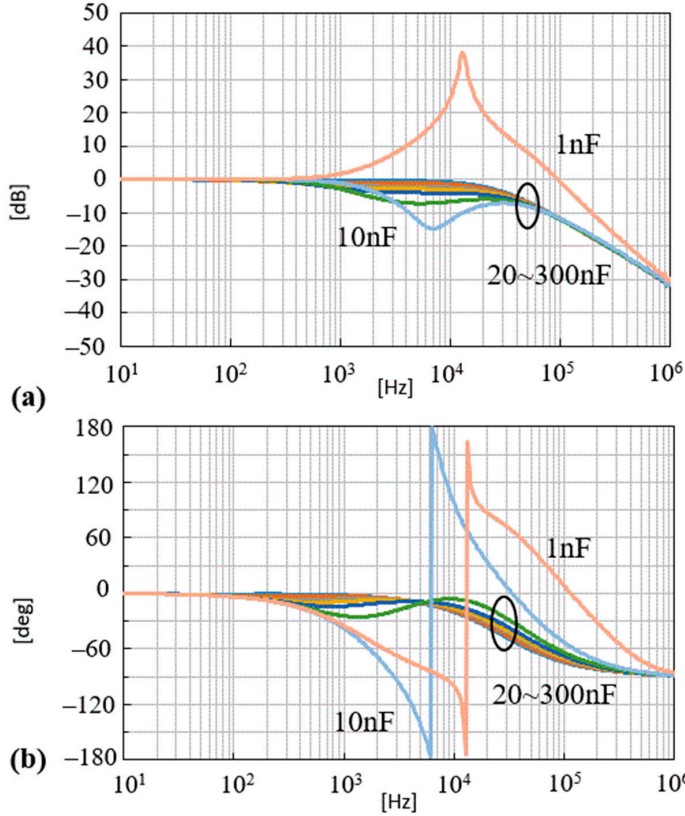

**Figure 6.** (**a**) Idea of a built-in POR and (**b**) the simulated waveform.

**Figure 7.** Bode plots with $C_{VDD}$ varied. (**a**) Gain vs. freq.; (**b**) Phase vs. freq.

## 3. Experiments

The proposed AC-DC converter was designed and fabricated in 65 nm low-Vt CMOS, as shown in Figure 8. The majority block was BGR in terms of area. The entire area was 0.081 mm$^2$, which is so small that it can be integrated into the same IoT IC chip. The pulse generator available at the lab only generated an AC peak of 10 V. As a result, the AC-DC converter was measured with a 100 kΩ resistor connected with the AC voltage source to an input power larger than 10 μW. A 6 μF capacitor was connected with V$_{DD}$.

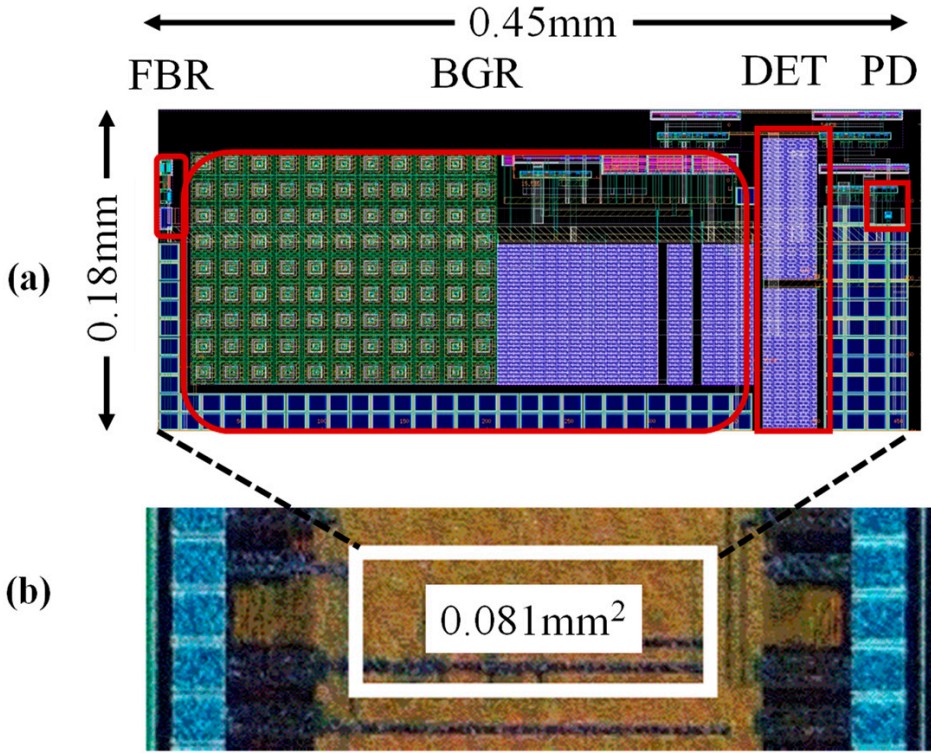

**Figure 8.** (**a**) Layout of the AC-DC converter and (**b**) a die photo.

Figure 9a shows I$_{DD}$ vs. V$_{DD}$ by varying the load resistance R$_L$. The AC-DC converter regulates V$_{DD}$ at 1 V with 10 μA or below. Figure 9b shows the measured waveform at different temperatures. The voltage source starts at 0.1 s with an amplitude of 10 V at 100 Hz. The converter charges up the load capacitance C$_{VDD}$ for 0.25 s until V$_{DD}$ reaches about 1 V. The measured average and ripple of V$_{DD}$ are summarized in Table 2.

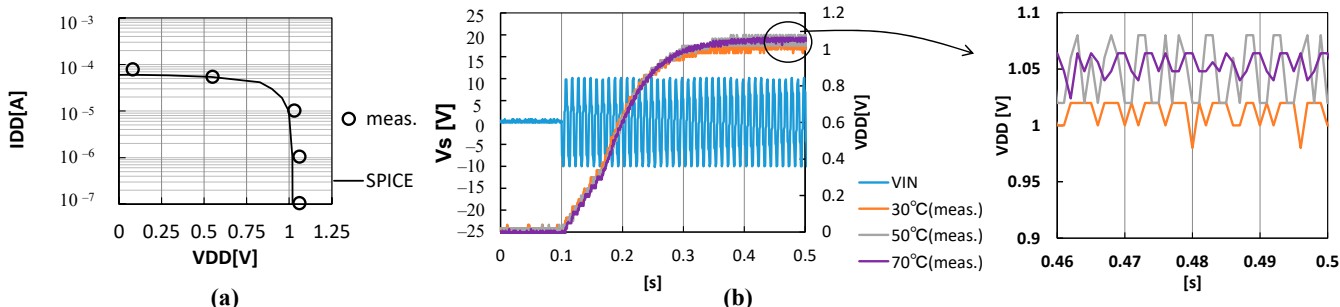

**Figure 9.** Measured I$_{DD}$ vs. V$_{DD}$ (**a**) and waveform (**b**).

**Table 2.** Measured $V_{DD}$ vs. temperature.

| Temp. | VDD |
|---|---|
| 0 °C | 1.00 V $\pm$ 20 mV |
| 27 °C | 1.05 V $\pm$ 30 mV |
| 70 °C | 1.05 V $\pm$ 22 mV |

To verify the effect of the built-in POR, measurements were also performed by connecting the source and drain of $P_{PD}$. As shown by "$V_{DD}$ without PMOS" in Figure 10, $V_{DD}$ was stuck at the ground level.

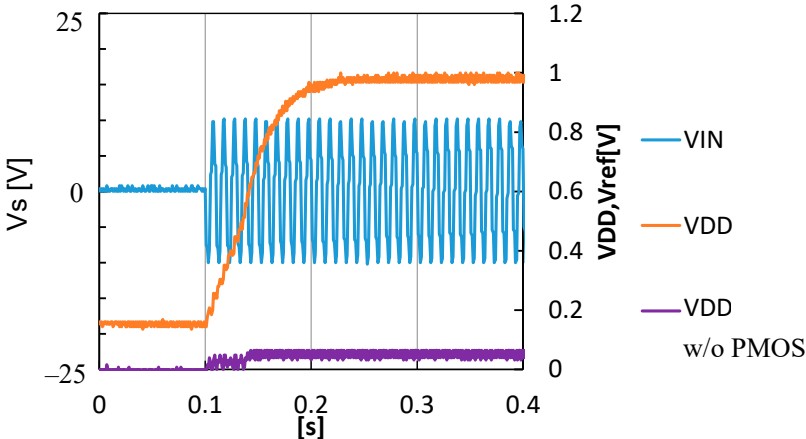

**Figure 10.** Measured waveform with and without $P_{PD}$.

## 4. Discussions

### 4.1. Comparison with Previously Reported Converters

Circuit features are compared in Table 3. In [6], the AC-DC converter was composed of FBR and a buck DC-DC converter. To allow for a high voltage input of 60 V at the peak, a BCD process was used, which provided 60 V transistors. Discrete diodes for FBR were required in addition to a converter chip. A high power efficiency of 85% was realized by the buck converter. In [10], another AC-DC converter was presented, which was composed of capacitor divider, switched capacitor converter, and FBR to convert power from 120 V mains. Wiring capacitors were used to manage a high voltage of 168 V without adding extra process steps or devices. As capacitance density was quite low, the converter size needed to be as large as 9.8 mm$^2$. In [11], another AC-DC converter was proposed to generate a standard CMOS-compatible voltage of 2 V from the magnetostrictive energy harvester (MS-EH) with a peak open circuit voltage of 0.5 V. Due to the on-chip oscillator running at 5 MHz to drive a charge pump circuit (CP), the control circuit consumed power of 18 μW. In [12], a DC-DC charge pump was developed for piezo-electric energy harvesting. The voltage conversion ratio was varied for energy efficient power conversion according to $V_A$. Area per maximum output power was realized with 3.1 [mm$^2$/mW]. On the other hand, a shunt regulator was used instead of the buck converter in this work at the cost of a reduction in PCE. However, when the transducer can generate sufficient power for the IoT chip even with the AC-DC converter with 43% PCE, it can be integrated into the same IoT chip without additional discrete components and the buck converter chip. Area per maximum output power was realized with 8.1 [mm$^2$/mW] in measurement and 0.81 [mm$^2$/mW] in simulation under the conditions of $R_S$ = 100 kΩ, $R_L$ = 10 kΩ, and $V_A$ = 30 V.

**Table 3.** Comparison table with previously reported converters.

| | Stanzione [6] | De Pelecijn [10] | Kawauchi [11] | Chen [12] | This Work |
|---|---|---|---|---|---|
| Energy source | ES-EH | AC mains | MR-EH | PZ-EH | ES-EH |
| Voltage conversion: Up or Down? | Down | Down | Up | Up/Down | Down |
| Architecture | FBR and Buck | Cap-div, SC, and FBR | FBR and AC-DC CP | FBR and DC-DC CP | FBR and Shunt |
| External components | FBR (4 diodes)/LCR (1L,1C,2R) | None (except for $C_{VDD}$) | | | |
| CMOS | 0.25 µm 60 V BCD and 3 V CMOS | 0.35 µm 12 V HV-CMOS | 65 nm 1 V low-Vt CMOS | 0.18 µm CMOS | 65 nm 1 V low-Vt CMOS |
| VDD regulation | N. A. | N. A. | No regulation | N. A. | ±5% |
| Control power | 500 nW | 50 nW | 18 µW | 4 µW (*1) | 700 nW |
| Maximum input peak voltage | 60 V | 168 V | 1 V | N. A. | 10 V (measured), 100 V (potentially) |
| Input power | 1 µW–1 mW | 20 µW | 22 µW | N.A. | 1 µW–100 µW |
| Output power | 1 µW–1 mW | 20 µW | 4 µW | 0.5–64 µW | 1 µW–100 µW |
| Power conversion efficiency | 85% | 81% | 23% | 72% (*1) | 43% |
| Die/circuit area | BCD (3 mm$^2$) and CMOS (N.A.) | 9.8 mm$^2$ | 0.11 mm$^2$ | 0.2 mm$^2$ | 0.081 mm$^2$ |
| Area [mm$^2$]/Max. output power [mW] | 4.6 | 612.5 | 27.5 | 3.1 | 8.1 (meas.), 0.81 (sim.) (*2) |

(*1) The data were taken from the condition of $V_{OUT}$ = 2 V and $V_A$ = 3 V. (*2) The data were simulated under the conditions of $R_S$ = 100 kΩ, $R_L$ = 10 kΩ, and $V_A$ = 30 V.

### 4.2. Limination of the Proposed Converter on $V_A$ and $R_S$ of ES-EH

To see which electrical parameters of ES-EH allowed the converter to output a regulated voltage of 1 V, SPICE simulation was performed with various amplitude voltages $V_A$ and output resistances $R_S$ of ES-EH under the condition of a load resistance of 1 MΩ, as shown in Figure 11. The middle area in blue shows that $V_{DD}$ is regulated between 0.95 V and 1.0 V. ES-EH with too high Rs inputs insufficient power into the regulator against the output power, whereas ES-EH with too low Rs injects too much power so that the PD path cannot pull down to the target regulation point. The lower bound on Rs can be reduced with a larger PD size.

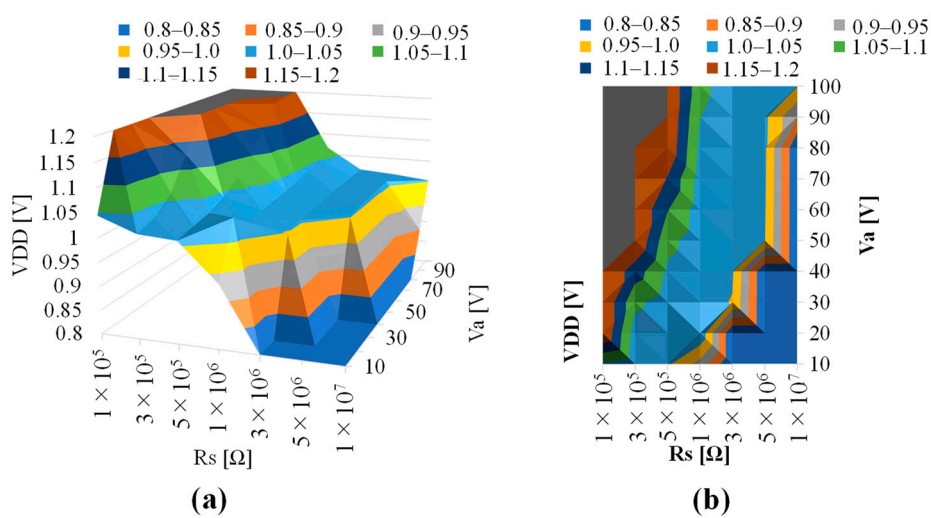

**Figure 11.** Simulated VDD with different $V_A$ and $R_S$ under a load resistance of 1 MΩ in (**a**) 3D and (**b**) top view plots.

## 5. Conclusions

This paper proposed an AC-DC converter which does not require external components for rectification and power conversion. It can be integrated in the same IoT chip with a small overhead area of 0.1 mm$^2$. This converter can provide a better option for electrostatic energy harvesting where the cost is the highest priority.

**Author Contributions:** Conceptualization, T.T.; methodology, Y.I. and T.T.; software, Y.I.; validation, Y.I. and T.T.; formal analysis, Y.I. and T.T.; investigation, Y.I. and T.T.; writing—original draft preparation, Y.I.; writing—review and editing, T.T.; funding acquisition, T.T. Both authors have read and agreed to the published version of the manuscript.

**Funding:** This research received no external funding.

**Acknowledgments:** This work is supported by d-lab.VDEC, Synopsys, Inc., Cadence Design Systems, Inc. Rohm Corp. and Micron Foundation. The authors wish to thank Associate M. Futagawa, H. Hirano and S. Ota for technical discussion.

**Conflicts of Interest:** The authors declare no conflict of interest.

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
