# Peer review of "A Fully Integrated AC-DC Converter in 1 V CMOS for Electrostatic Vibration Energy Transducer with an Open Circuit Voltage of 10 V"

_electronics, doi:10.3390/electronics10101185_

Round 1

Reviewer 1 Report

Below you can find some examples of remarks and suggestions that were made while reading the article:

Remarks:

The line 36: typo in the word “resistor”.

The line 104: Adjust heading of the section 3 to the left. Bold the font of the heading.

Suggestions:

Remember to use references when stating equations.

This paper mainly presents an AC-DC converter for electrostatic vibration energy harvesting. This makes the topic very interesting for Energy Harvesting Systems.

The introduction provides a good, generalized background of the topic that quickly gives the reader an appreciation of modern power electronics.

I think the motivations presented in the paper are clear and not in dispute.

The paper fully describes the posed question.

The experimental circuit is quite modern and very suitable for this study. The circuit was fabricated in 65nm low-Vt CMOS. 1V regulation was measured successfully at 20-70oC with power conversion efficiency of 43%. The proposed converter can provide a better option for electrostatic energy harvesting where the cost is the  highest priority.

The literature cited  in this article is relevant to the study, but more citations (references) throughout the text would be useful.

This paper is well written. The text for reader is  clear and easy to read.

The conclusions in the paper are consistent with the evidence and argument presented. These conclusions address  the main question posed.

The work is done at a good level and can be recommended after minor revision for publication.

Author Response

The authors wish to thank you for providing valuable comments and suggestions. Can you please check the uploaded file?

Reviewer 2 Report

Paper presents AC-DC converter realized in 65nm low-Vt CMOS technology, for potential application in energy harvesting.

Comment are listed bellow:

  1. It is not usual to refer to figures from other articles (Ref [6] ) especially in Introduction section (Figure 1). Figures from other articles could be included in the authors’ paper only if it is needed to explain some specific issues of current paper which is not the case here. Do authors have permission and copyright to distribute this figure. I highly recommend to remove Figure 1 from manuscript.

  1. Paper has weak literature review. Authors should review more articles dealing with fully integrated AC-DC converters and compare their solution to existing state of the art solutions.

  1. Authors should empathise the novelty of their work if any.

  1. Lines in Figure 10 referring to different temperature conditions could not be distinguished. I suggest introducing some symbols with lines to improve visibility.

  1. Section 4, comparison with one reference [6] is not sufficient. Please include in Table III comparison with other AC-DC converters available in literature.

  1. Vibration and energy harvesting are mentioned only in title of the paper, introduction part and conclusion. Discussion on this topic with concrete results should be included in the main part of manuscript.

  1. Authors claim in introduction part that the priority of their converter in cost and form factor. Paper lacks comparison of the cost and form factor with available solutions in literature, otherwise it could not be listed as the advantage of realized convertor.

Author Response

(The authors gave the same response as above.)

Round 2

Reviewer 2 Report

As indicated in previous report nor the cost nor the form factor advantages issues (mentioned in introductory and conclusion part) are not explained and compared with other similar converters.

Included sentence " Thus, the form factor and the
 cost of the edge devices can be reduced theoretically." must be supported by scientific explanation.

Literature review is still week authors should survey more articles and include them in their manuscript. Also most of the  references are obsolete (9/11 are from year 2015 or before). Authors should review recent sources from the literature in order to support up-to-date of their work.

Author Response

Thanks for your feedback. Please see the attachment. 

Round 3

Reviewer 2 Report

Authors have improved previous version of manuscript.